

# High throughput sequencing-based analysis of the soil bacterial community structure and functions of Tamarix shrubs in the lower reaches of the Tarim River

Fangnan Xiao, Yuanyuan Li, Guifang Li, Yaling He, Xinhua Lv, Li Zhuang and Xiaozhen Pu

College of Life Science, Shihezi University, Shihezi, Xinjiang Uygur Autonomous Region, China

## ABSTRACT

Tamarix is a dominant species in the Tarim River Basin, the longest inland river in China. Tamarix plays an important role in the ecological restoration of this region. In this study, to investigate the soil bacterial community diversity in Tamarix shrubs, we collected soil samples from the inside and edge of the canopy and the edge of nebkhas and non-nebkhas Tamarix shrubs located near the Yingsu section in the lower reaches of Tarim River. High throughput sequencing technology was employed to discern the composition and function of soil bacterial communities in nebkhas and non-nebkhas Tamarix shrubs. Besides, the physicochemical properties of soil and the spatial distribution characteristics of soil bacteria and their correlation were analyzed. The outcomes of this analysis demonstrated that different parts of Tamarix shrubs had significantly different effects on soil pH, total K (TK), available K (AK), ammonium N ($NH_4^+$), and available P (AP) values ($P < 0.05$), but not on soil moisture (SWC), total salt (TDS), electrical conductivity (EC), organic matter (OM), total N (TN), total P (TP), and nitrate N ($NO_3^-$) values. The soil bacterial communities identified in Tamarix shrubs were categorized into two kingdoms, 71 phyla, 161 classes, 345 orders, 473 families, and 702 genera. Halobacterota, unidentified bacteria, and Proteobacteria were found to be dominant phyla. The correlation between the soil physicochemical factors and soil bacterial community was analyzed, and as per the outcomes OM, AK, AP, EC, and $NH_4^+$ were found to primarily affect the structure of the soil bacterial community. SWC, TK and pH were positively correlated with each other, but negatively correlated with other soil factors. At the phyla level, a significantly positive correlation was observed between the Halobacterota and AP, OM as well as Bacteroidota and AK ($P < 0.01$), but a significantly negative correlation was observed between the Chloroflexi and AK, EC ($P < 0.01$). The PICRUSt software was employed to predict the functional genes. A total of 6,195 KEGG ortholog genes were obtained. The function of soil bacteria was annotated, and six metabolic pathways in level 1, 41 metabolic pathways in level 2, and 307 metabolic pathways in level 3 were enriched, among which the functional gene related to metabolism, genetic information processing, and environmental information processing was found to have the dominant advantage. The results showed that the nebkhas and canopy of Tamarix shrubs had a certain

Corresponding author
Xiaozhen Pu, 3425825446@qq.com

![PeerJ]

enrichment effect on soil nutrients content, and bacterial abundance and significant effects on the structure and function of the soil bacterial community.

# INTRODUCTION

The Tarim River is the longest inland river in China. Ecosystem stability in the Tarim River Basin could be attributed to the riparian desert vegetation. Tamarix is a commonly occurring shrub in the Tarim River Basin of Xinjiang, China. Besides, the Tamarix plant is resistant to drought, salt, and wind (*Liu, 1996*; *Li et al., 2007*), and it plays a vital role in sand fixation and windbreak (*Yang et al., 2008*). Sand around the Tamarix shrubs can gather to form Tamarix nebkhas (*Yang et al., 2012*). These nebkhas are also referred to as "fertilizer islands" (*Nickling & Wolfe, 1994*) and they increase the nutrient content of the arid desert soil.

Soil microorganisms are a vital component of the soil ecosystem (*Girvan et al., 2010*). Soil microbes decompose the shrub litter and participate in nutrient cycling and energy flow of the soil ecosystem (*Vasconcellos et al., 2013*; *Preem et al., 2012*), and thus promote the "fertile island" development in the nebkhas formation (*Wei et al., 2013*; *Cao et al., 2016*). Metabolic activities of soil microbes exacerbate soil respiration, which increases ecosystem productivity (*Bauhus, Pare & Cote, 1998*) and thus promotes plant growth and development. Bacteria are the most abundant group of soil microbes with the highest genetic diversity (*John et al., 2002*). The community structure diversity and species richness of soil bacteria could reflect the quality and fertility of soil to a certain extent (*Hu et al., 2001*). Previous studies have investigated the microbial composition of Tamarix nebkhas in the arid area of Tarim River using plate cultivation methods. As per the outcomes of these studies, abundances of soil microbes vary inside and outside of Tamarix nebkhas canopy (*Chen et al., 2008a*). The nebkhas shrub showed an enrichment effect on microbial abundances (*Chen et al., 2008b*). However, the soil microbe's distribution characteristics at different taxonomic levels in Tamarix shrubs remain elusive. High throughput sequencing technology can comprehensively and accurately analyze the characteristics of soil microbial composition and functional diversity (*Xia & Jia, 2014*). The relevant literatures of high-throughput sequencing mainly focus on the study of forest (*Deng et al., 2020*), wetland (*Zou et al., 2019*) and grassland (*Wu et al., 2019*) ecosystem, but the study of desert shrubs in arid area is rarely reported. Based on previous studies, using high-throughput sequencing technology can more comprehensively analyze the community structure and function of soil microorganisms in Tamarix shrubs and can provide scientific basis for renewing soil microbial resources of Tamarix shrubs in arid area.

The Tarim River Basin area has the most concentrated distribution of Tamarix in China (*Chen et al., 2004*; *Yang et al., 2002*). As the dominant species, Tamarix plays an important

role in stabilizing the fragile ecosystem of the Tarim River Basin. In this study, high throughput sequencing technology was employed to analyze the soil bacterial community composition in the nebkhas and non-nebkhas Tamarix shrubs located in the lower reaches of the Tarim River. Also, the distribution of soil microbes in Tamarix shrubs at different taxonomic levels and effects of Tamarix nebkhas and soil factors on soil bacterial community and function were investigated. This study aimed to provide basic scientific data for the ecological restoration of vegetation in the Tarim River Basin.

# MATERIALS AND METHODS

## Site description and sampling

The study site was located near the Yingsu section in the lower reaches of Tarim River (with the geographical coordinates of 40°28′ N, 87°51′ E and 850 m elevation). The study area has a typical extreme arid temperate continental climate with low rainfall and strong evaporation. It receives an annual average rainfall of 17.4–42.0 mm, and its mean annual evaporation is 2,671.4–2,902.2 mm. During the study, the average annual temperature of the study site was 10.6–11.5 °C, and it was hot in summer and cold in winter. The average temperature in July was 20–30 °C. In the span of 30 ~ 40 days, temperature crossed 35 °C and the highest temperature recorded was 43.6 °C. The average temperature in January ranged from −10 °C to −20 °C, and the lowest temperature recorded was −27.5 °C. The duration of sunshine was around 2,780–2,980 h. The annual solar radiation was 5,692–6,360 MJ·m$^{-2}$, the frost-free period ranged from 187 to 214 days, and the maximum wind speed was around 20–40 m/s (*Yang & He, 2000*). According to our fieldwork, the study site was primarily populated with *Populus euphratica* Oliv trees, *Tamarix* spp, *Halostachys capsica* shrubs, and *Alhagi sparsifolia Shap. ex kell*, *Glycyrrhiza uralensis Fisch*. herbs. Tamarix was found to be the dominant shrub species in this area.

   We sampled soil from the Tamarix plant community area near the Yingsu section of the lower reaches of the Tarim River in September 2020. Three nebkhas Tamarix shrubs (X) and three non-nebkhas Tamarix shrubs (F) with almost the same size were selected from the sample plots (three shrubs were treated as three replicates). The soil samples were collected radially from the (a) inside of the canopy, (b) the edge of the canopy, and (c) the edge of the shrubs from the inside to the outside based on the distance from the base of the plant (the spacing of each position was approximately one m) (Fig. S1). For soil sampling, firstly, extremely arid topsoil of 0–20 cm was removed, and four soil samples from 20–40 cm soil depth from the east, south, west, and north directions at each position of each shrub were collected. These soil samples were mixed, and excess soil sample was removed by using the quartering method. Thus, a total of 18 soil samples were obtained, and each sample was divided into three parts. The first part was placed in an icebox, and the fresh weight of the soil was immediately weighed. The soil samples were immediately transferred to the laboratory and dried using the oven to determine the soil water content. The second part was put into a sealed bag and air-dried in the laboratory, and then sieved through two mm sieve to determine the physical and

chemical properties of the soil. The third part was placed in a centrifuge tube and stored in a liquid nitrogen tank for DNA extraction.

## Soil analysis

Soil water content (SWC) was measured using the oven-dry method. Soil chemical properties were determined by the method described by *Bao (2000)*. Organic matter (OM) content was determined using the potassium dichromate volumetric external heating method. Total nitrogen (TN) content was determined by employing the perchlorate-sulfuric acid digestion method (1,035 automatic nitrogen determination apparatus, FOSS). Total phosphorus content (TP) was determined by the acid solution-molybdenum antimony colorimetric method (CARY60 UV-Vis spectrophotometer, Agilent). Total potassium (TK) was measured using the acid dissolution-atomic absorption method and atomic absorption spectrometer (Thermo Fisher Scientific, Waltham, MA, USA). Nitrate nitrogen ($NO_3^-$) and ammonium nitrogen ($NH_4^+$) were determined using 0.01 M calcium chloride extraction method and flow analyzer (BRAN+LUEBBE AA3). Available phosphorus (AP) was determined by using sodium bicarbonate extraction-molybdenum antimony colorimetric method and CARY60 UV-Vis spectrophotometer (Agilent, Santa Clara, CA, USA). Available potassium (AK) was determined by ammonium acetate extraction-atomic absorption method and atomic absorption spectrometer (Thermo Fisher Scientific, Waltham, MA, USA). pH was determined using the Mettler-Toledo FiveEasy Plus pH meter. Electrical conductivity (EC) was measured by Anna HI2315 conductivity meter. The dry-residue method was employed to determine the total dissolved salts (TDS).

## DNA extraction, amplification, and sequencing of soil bacteria

CTAB method was employed to extract genomic DNA of soil samples. The purity and concentration of DNA were detected by agarose gel electrophoresis. An appropriate amount of DNA sample was diluted to one ng/μl using sterile water. The diluted genomic DNA was used as a template, and the 16S V4 variable region was PCR amplified using specific primers with barcode (515F-GTGCCAGCMGCCGCGGTAA and 806R-GGACTACHVGGGTWTCTAAT), Phusion® high-fidelity PCR Master Mix with GC Buffer (New England Biolabs, Ipswich, MA, USA), and high-efficiency high-fidelity enzyme. To ensure amplification efficiency and accuracy, PCR products were checked on 2% agarose gel electrophoresis. The target bands were recovered using the gel recovery kit (Qiagen, Hilden, Germany). Truseq® DNA PCR-Free Sample Preparation Kit was used for library construction. The constructed libraries were analyzed using Qubit and Q-PCR and then sent to Beijing Compson Biotechnology Co. Ltd for computerized sequencing using NovaSeQ 6000 system.

## Sequence analysis

According to the barcode sequence and PCR amplification primer sequence, each sample data was split from offline data. After cutting the barcode and primer sequence, FLASH (*Magoč & Salzberg, 2011*) was used to splice each sample's reads, and the spliced

sequence were Raw Tags. The Raw Tags was filtered using QiIME version 1.9.1 software (*Caporaso et al., 2010*). The chimeric sequences were removed (*Rognes et al., 2016*), and the Effective Tags were obtained. UPARSE (*Haas et al., 2011*) software version 7.0.1001 was used to cluster all the Effective Tags of all samples. The sequences were divided into operational taxonomic units (OTUs) with 97% identity. Furthermore, the Mothur method was employed to select the representative sequence of OTUs. To obtain taxonomic information and calculate the community composition of each sample at each taxonomic level, species annotation analysis was carried out using the SSSurrNA database of SILVA 138 (*Edgar, 2013*; *Wang et al., 2007*).

## Data analysis

Alpha diversity analysis, including Shannon, Simpson, Chao1, and ACE indices, was performed using QIIME software. The UniFrac distance was calculated using QIIME software, and the unweighted pair-group method with arithmetic means (UPGMA) cluster tree was constructed. NMDS and db-RDA diagrams were drawn using the Vegan software package of R software version 2.15.3. Linear Discriminant Analysis Effect Size (LEfSe) analysis was performed by employing LEfSe software (*Segata et al., 2011*). Prediction of bacterial community function was performed by using the PICRUSt software (*Langille et al., 2013*) and comparing it with the Greengene database. One-way ANOVA was used to analyze the differences of soil factors and microbial diversity in different locations of nebkhas Tamarix shrubs and non-nebkhas Tamarix shrubs. Sample-paired T test was used to analyze soil factors and soil microbial diversity between nebkhas Tamarix shrubs and non- nebkhas Tamarix shrubs.

## RESULTS

### Analysis of soil physical and chemical properties of Tamarix shrubs

The soil sampled in the study site was weakly alkaline with 7.5–8.0 pH. The pH of nebkhas Tamarix shrubs was significantly higher than that of non-nebkhas Tamarix shrubs, but pH within the shrub did not vary significantly. The maximum values of TDS, TP, TN, AK, AP, and $NO_3^-$ was observed in the samples from inside of nebkhas and non-nebkhas Tamarix shrub' canopy (Xa and Fa); besides, the AK content was significantly higher than other soil nutrients, and it decreased gradually from inside to outside in the shrubs. Besides, AK content differed significantly in the nebkhas Tamarix shrubs ($P < 0.05$), but it did not differ significantly between the nebkhas and non-nebkhas Tamarix shrubs. The TK content in Tamarix nebkhas shrubs was significantly higher than that in non-nebkhas Tamarix shrubs ($P < 0.05$). The TK content differed significantly at different position of non-nebkhas Tamarix shrubs ($P < 0.05$). $NH_4^+$ value was highest in Fc, and it was significantly higher than other position in the shrub ($P < 0.05$). The $NH_4^+$ content was significantly higher in the non-nebkhas shrubs than in the nebkhas shrubs ($P < 0.05$). Besides, AP was significantly higher inside the canopy than outside the canopy in shrubs ($P < 0.05$), but no significant difference was observed between nebkhas and non-nebkhas Tamarix shrubs. OM, TN, TP, and $NO_3^-$ did not differ significantly between Tamarix shrubs (Table 1).

**Table 1 Analysis of soil physical and chemical properties of Tamarix shrubs.**

| | Shrubs-position | Nebkhas Tamarix shrubs | | | | Non-nebkhas Tamarix shrubs | | | |
|---|---|---|---|---|---|---|---|---|---|
| | | Inside canopy (Xa) | Canopy edge (Xb) | Shrubs edge (Xc) | Average | Inside canopy (Fa) | Canopy edge (Fb) | Shrubs edge (Fc) | Average |
| Soil-factors | OM(g · kg-1) | 9.70 ± 1.84a | 6.06 ± 0.58a | 5.22 ± 1.68a | 6.99 ± 1.01A | 7.63 ± 0.54a | 7.01 ± 0.31a | 8.76 ± 0.28a | 7.80 ± 0.32A |
| | TN(g · kg-1) | 0.55 ± 0.06a | 0.41 ± 0.03a | 0.38 ± 0.09a | 0.45 ± 0.04A | 0.50 ± 0.09a | 0.41 ± 0.04a | 0.46 ± 0.05a | 0.47 ± 0.03A |
| | TP(g · kg-1) | 0.57 ± 0.02a | 0.53 ± 0.02a | 0.49 ± 0.02a | 0.53 ± 0.01A | 0.56 ± 0.01a | 0.54 ± 0.01a | 0.55 ± 0.01a | 0.55 ± 0.01A |
| | TK(g · kg-1) | 18.35 ± 0.59a | 18.56 ± 0.75a | 18.48 ± 0.13a | 18.47 ± 0.28A | 17.18 ± 0.08b | 18.03 ± 0.07a | 17.92 ± 0.21a | 17.71 ± 0.14B |
| | $NO_3^-$(mg·kg-1) | 6.78 ± 3.72a | 4.77 ± 1.76a | 4.63 ± 1.72a | 5.39 ± 1.33A | 7.05 ± 2.90a | 4.91 ± 1.43a | 5.08 ± 0.24a | 5.68 ± 0.99A |
| | $NH_4^+$(mg·kg-1) | 4.12 ± 0.69a | 3.25 ± 0.35a | 3.44 ± 0.34a | 3.61 ± 0.27B | 4.11 ± 0.21b | 3.57 ± 0.51b | 12.15 ± 0.28a | 6.61 ± 1.39A |
| | AP(mg·kg-1) | 3.95 ± 0.73a | 1.72 ± 0.44b | 1.36 ± 0.61b | 2.34 ± 0.50A | 4.08 ± 0.60a | 1.78 ± 0.29b | 3.68 ± 0.71ab | 3.18 ± 0.45A |
| | AK(mg·kg-1) | 679.17 ± 133.61a | 316.74 ± 46.72b | 186.08 ± 41.54b | 394.00 ± 85.15A | 406.87 ± 101.73a | 387.01 ± 93.98a | 217.97 ± 10.31a | 337.29 ± 50.05A |
| | pH (1:5) | 7.94 ± 0.07a | 7.71 ± 0.06a | 8.01 ± 0.25a | 7.88 ± 0.08A | 7.72 ± 0.04a | 7.53 ± 0.07a | 7.52 ± 0.02a | 7.59 ± 0.04B |
| | EC(ms·cm-1) | 7.37 ± 0.33a | 4.48 ± 0.44a | 4.06 ± 1.18a | 5.31 ± 0.64A | 5.71 ± 0.34a | 5.96 ± 1.32a | 5.64 ± 0.34a | 5.77 ± 0.41A |
| | TDS(g · kg-1) | 26.76 ± 2.03a | 15.75 ± 1.07a | 14.21 ± 4.82a | 18.91 ± 2.50A | 20.81 ± 0.73a | 20.13 ± 5.20a | 18.17 ± 1.21a | 19.71 ± 1.60A |
| | SWC(%) | 0.56a±0.06a | 1.62 ± 0.53a | 2.84 ± 1.33a | 1.67 ± 0.53A | 1.44 ± 0.27a | 1.81 ± 0.09a | 1.05 ± 0.01a | 1.43 ± 0.13A |

Note:
The data are average ± standard error, different lower-case letters after the same line mean significant difference in different positions in the shrub ($P < 0.05$), different capital letters after the same line mean significant difference of the shrub with and without nebkhas ($P < 0.05$).

**Table 2 Bacterial diversity and abundance in soil from Tamarix shrubs.**

| Shrubs-position | Nebkhas Tamarix shrubs | | | | Non-nebkhas Tamarix shrubs | | | |
|---|---|---|---|---|---|---|---|---|
| | Inside -canopy (Xa) | Canopy-edge (Xb) | Shrubs- edge (Xc) | Average (X) | Inside -canopy (Fa) | Canopy- edge (Fb) | Shrubs-edge (Fc) | Average (F) |
| shannon index | 8.03 ± 0.59a | 8.75 ± 0.15a | 8.51 ± 0.29a | 8.43 ± 0.22A | 8.56 ± 0.31a | 8.44 ± 0.45a | 8.63 ± 0.19a | 8.54 ± 0.17A |
| simpson index | 0.98 ± 0.01a | 0.99 ± 0.01a | 0.99 ± 0.01a | 0.99 ± 0.01a | 0.99 ± 0.01a | 0.98 ± 0.01a | 0.99 ± 0.01a | 0.99 ± 0.01A |
| chao1 index | 2381.02 + 710.78a | 2601.11 + 199.39a | 2582.78 ± 173.64a | 2521.63 ± 221.74A | 2841.77 ± 259.01a | 2648.93 ± 418.84a | 2646.91 ± 146.51a | 2712.54 ± 151.79A |
| ACE index | 2434.32 ± 727.51a | 2613.06 ± 199.33a | 2612.85 ± 179.67a | 2553.41 ± 225.82A | 2881.02 ± 261.12a | 2664.29 ± 423.98a | 2682.53 ± 149.72a | 2742.62 ± 154.06A |
| coverage | 0.993 | 0.993 | 0.993 | 0.993 | 0.992 | 0.993 | 0.993 | 0.993 |

Note:
The data are average ± standard error, different lower-case letters after the same line mean significant difference in different positions in the shrub ($P < 0.05$), different capital letters after the same line mean significant difference of the shrub with and without nebkhas ($P < 0.05$).

## Analysis of bacterial alpha diversity in the soil from Tamarix shrubs

The OTU coverage in all experimental groups was higher than 99%, indicating sufficient sequencing depth. Shannon, Simpson, Chao1, and ACE indices were not significantly different between nebkhas and non-nebkhas Tamarix shrubs, indicating no significant differences in the abundance, diversity, and evenness of bacterial community in the nebkhas and non-nebkhas Tamarix shrubs (Table 2).

The Venn diagram was constructed to demonstrate the number of unique and common OTUs among different groups and for the pictorial representation of the difference in the

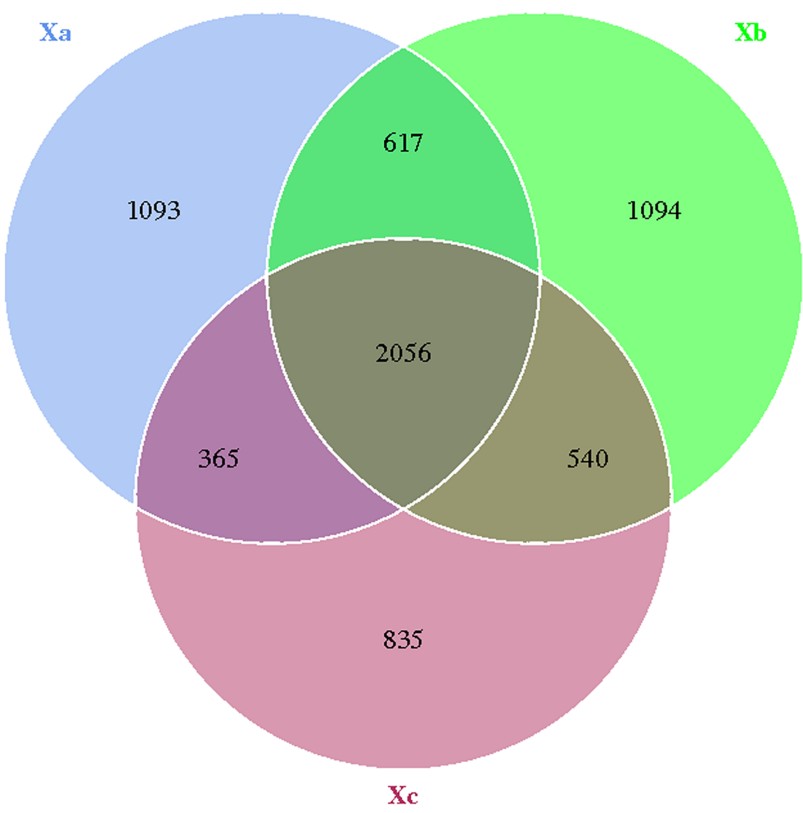

**Figure 1 Venn diagram of the distribution of soil bacteria OTUs in the nebkhas Tamarix shrubs.**

number of species among different groups. The number of bacterial OTUs shared by three groups in nebkhas Tamarix shrubs was 2,056, which accounted for 31.2% of the total OTUs (Fig. 1). Xa, Xb, and Xc contained 1,093, 1,094, and 835 unique OTUs, accounting for 16.6%, 16.6%, and 12.7% of the total OTUs, respectively. In the non-nebkhas Tamarix shrubs (Fig. 2), the number of OTUs shared by the three groups was 2205, accounting for 34.4% of the total OTUs. Fa, Fb, and Fc contained 989, 991, and 555 unique OTUs, accounting for 15.4%, 15.4%, and 8.6% of the total OTUs, respectively. In both nebkhas and non-nebkhas Tamarix shrubs, the number of OTUs inside and at the edge of the canopy was greater than at the edge of the shrubs. Besides, the number of OTUs in nebkhas Tamarix shrubs was greater than that in non-nebkhas Tamarix shrubs.

## Analysis of soil bacterial community composition at the phylum level

Based on the high throughput sequencing results, the number of OTUs that could be annotated to the database accounted for 99.8% of the total OTUs. Thus, a total of two kingdoms, 71 phyla, 161 classes, 345 orders, 473 families, and 702 genera were detected. The UPGMA analysis was used to analyze the differences in the bacterial community composition between the top 10 phyla and identify the highest abundant phyla (Fig. 3). Based on the average relative abundance, we found that dominant bacterial species belonged to Halobacterota, unidentified bacteria, and Proteobacteria, and these phyla
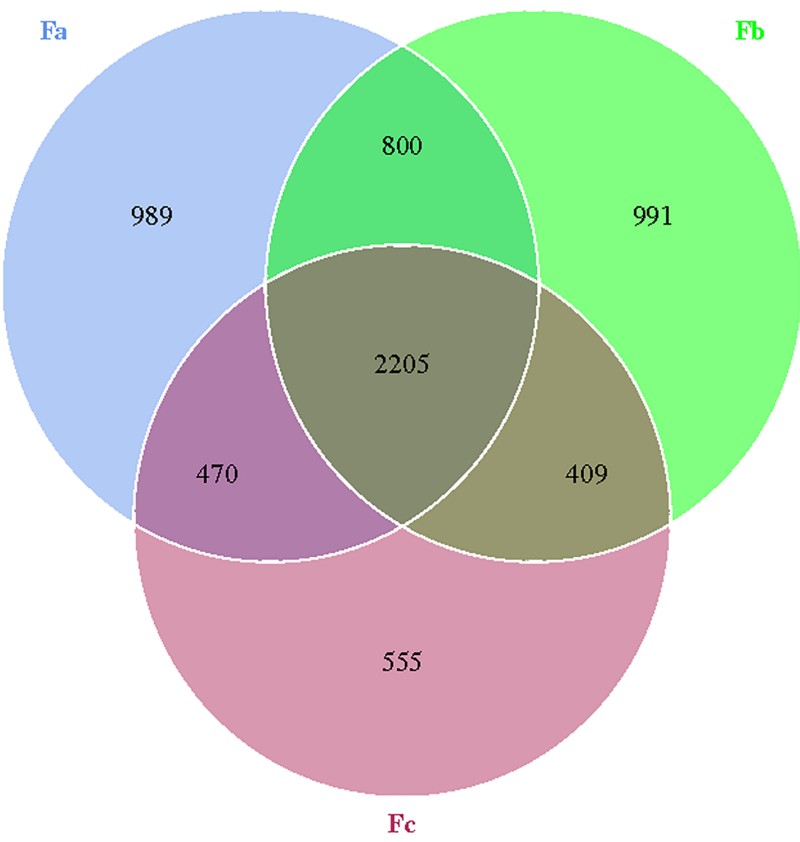

**Figure 2 Venn diagram of the distribution of soil bacteria OTUs in the non- nebkhas Tamarix shrubs.**

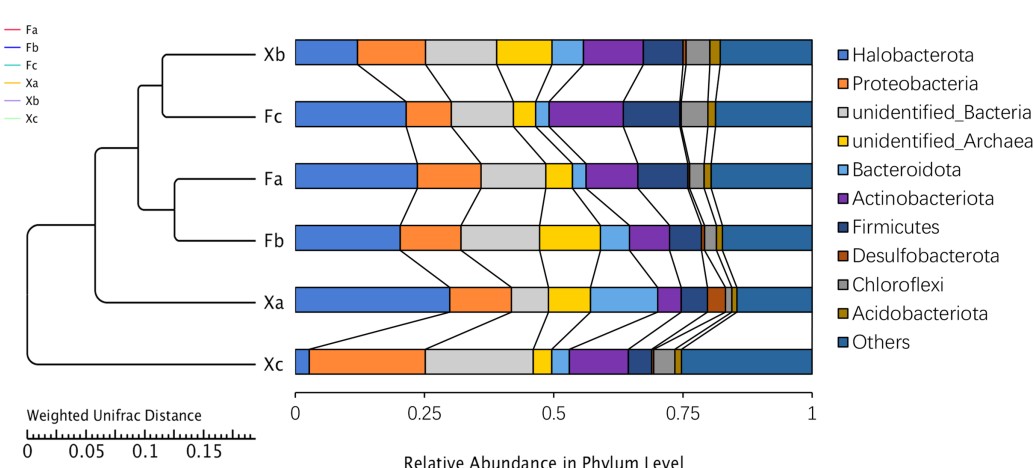

**Figure 3 UPGM cluster tree based on Weighted UniFrac distance at the phylum level of soil bacteria.**

accounted for 18.3%, 13.6%, and 13.4% of the average relative abundance, respectively. Halobacterota had the highest relative abundance in Xa, and Proteobacteria and unidentified bacteria had the highest relative abundance in Xc. The second most dominant phyla were Actinobacteriota, unidentified-Archaea, Firmicutes, Bacteroidota, and

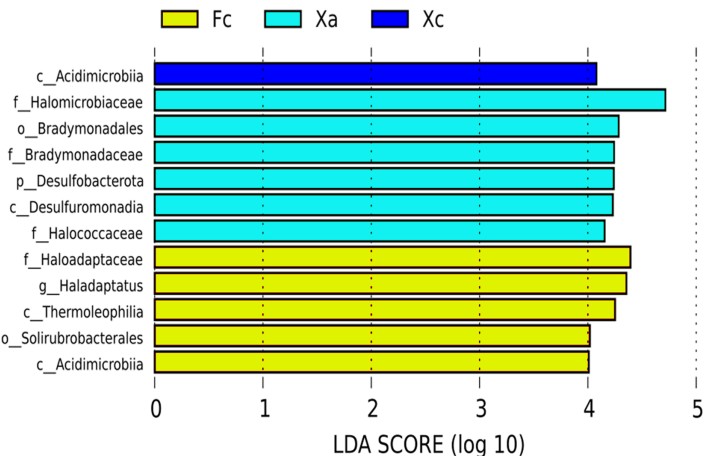

**Figure 4 Soil bacteria LDA value distribution histogram in Tamarix shrubs.**

Chloroflexi, accounting for 10.0%, 7.3%, 7.4%, 5.6%, and 3.3% of the average relative abundance of the total samples, respectively. Other dominant phyla were Acidobacteriota and Desulfobacterota, accounting for 1.4% and 1% of the average relative abundance, respectively. The remaining 61 phyla with a relative abundance of less than 1% were categorized as others, which accounted for 18.7% of the average relative abundance. As demonstrated in Fig. 3, Fa and Fb, Xb and Fc were clustered into one group. It indicated the highest similarity in the bacterial community composition between Fa and Fb; also, Xb and Fc. These four groups were clustered into one group and then clustered together with Xa and Xc, respectively, which demonstrated that the soil microbial community composition of Xa and Xc was different from these four groups. The clustering distance between Fa, Fb and Fc was closer than that between Xa, Xb and Xc indicated that the difference of bacterial community composition in nebkhas Tamarix shrubs was higher than non-nebkhas Tamarix shrubs.

## LEfSe analysis of different species of soil bacterial communities in Tamarix shrubs

LEFSE (Linear Discriminant Analysis Effect Size) is an analytical tool used to discover and interpret high-dimensional biomarkers (genes, pathways and taxon), which can be used to compare two or more groups. It emphasizes statistical significance and biological correlation, and can look for biomarkers with statistical differences between groups. In the LEfSe analysis of biomarker (signifcant differences species) from different groups in Tamarix shrubs, the LDA Score was set as four, and the species with LDA value > 4 were selected as the biomarker of the bacterial community in Tamarix shrubs. Based on the histogram of LDA value distribution (Fig. 4), a total of 12 biomarkers were identified, of which six, five, and one biomarker from different taxonomic levels belonged to Xa, Fc, and Xc groups, respectively. At the phylum level, only Desulfobacterota was enriched in the Xa group. At the class level, Desulfuromonadia was primarily enriched in Xa, Thermoleophilia and Actinobacteriota_Acidimicrobiia were enriched in Fc, and

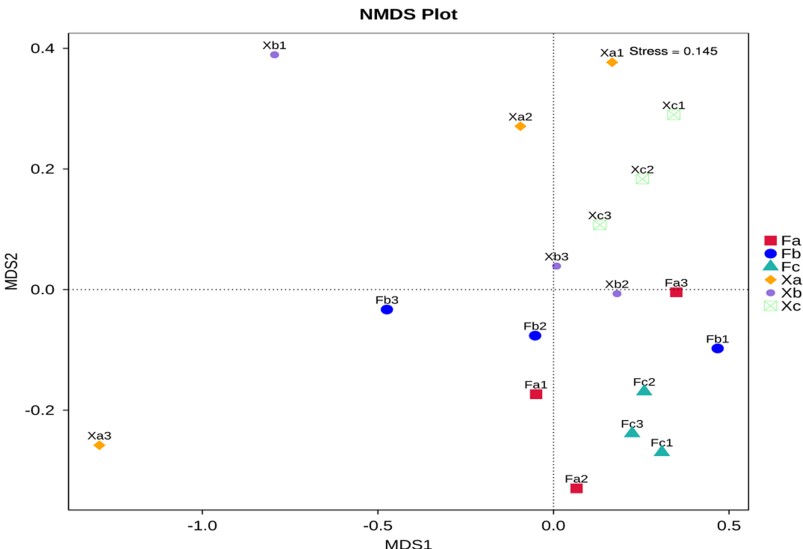

**Figure 5 NMDS analysis of soil bacterial community in Tamarix shrubs.**

unidentified bacteria and Acidimicrobiia were mainly enriched in Xc. At the order level, Bradymonadales was enriched in Xa and Solirubrobacterales in Fc. At the family level, Halomicrobiaceae, Bradymonadaceae, and Halococcaceae were mainly enriched in Xa, and Haloadaptaceae were mainly enriched in Fc. At the genus level, only Haladaptatus was enriched in Fc.

## NMDS analysis of soil bacterial community structure in Tamarix shrubs

Non-Metric Multi-Dimensional Scaling Analysis (NMDS) is a nonlinear mode of analysis based on Bray-Curtis distance, which signifies the nonlinear structure of data. NMDS can analyze the differences between samples in low dimensions. Each point in the figure represents a sample as well as samples in the same group are represented in the same color, and the difference becomes significant when the distance between the samples is large. A stress value of less than 0.2 is essential to obtain the NMDS analysis results. The NMDS analysis's stress value obtained in this study was 0.145, and thus it was used for subsequent analysis (Fig. 5). Soil samples of Tamarix nebkhas shrubs were distributed in the first, second, and third quadrants but concentrated only in the first quadrant. Soil samples of non-nebkhas Tamarix shrubs were distributed in the third and fourth quadrant and concentrated only in the fourth quadrant. These results indicated differences in bacterial community structure between nebkhas and non-nebkhas Tamarix shrubs. The distribution of soil samples of nebkhas Tamarix shrubs was loose, and that of non-nebkhas Tamarix shrubs was relatively concentrated. It indicated that the diversity of bacterial community structure in nebkhas Tamarix shrubs was greater than that in non-nebkhas Tamarix shrubs, which is in line with the results of the UPGMA cluster analysis above.

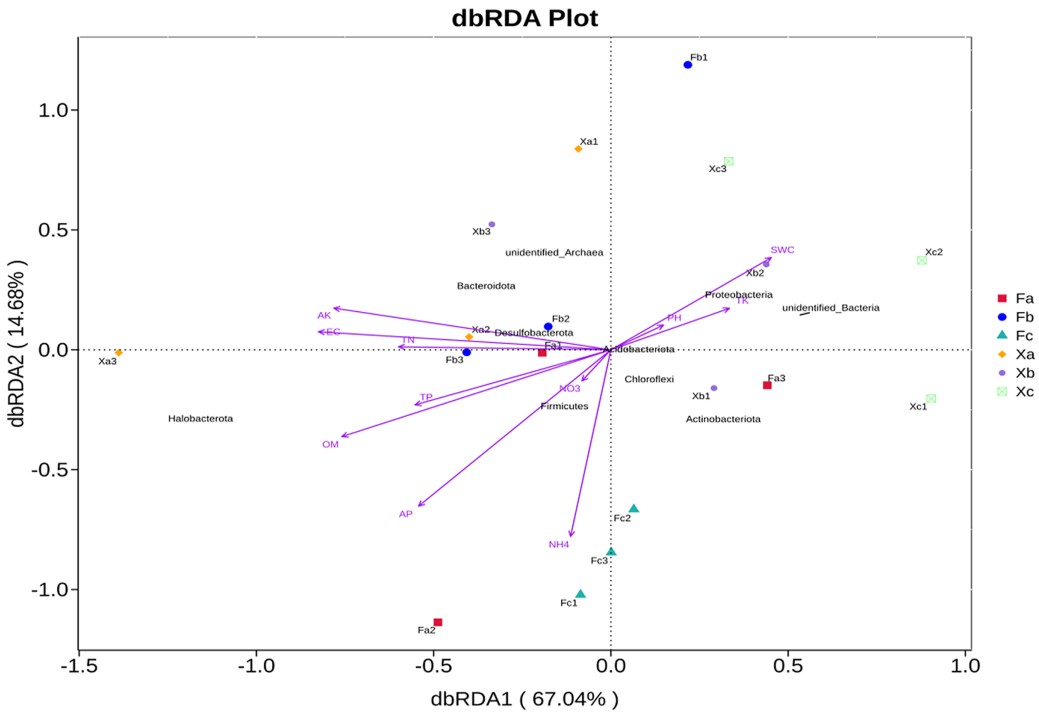

**Figure 6 db—RDA analysis of soil bacterial community in Tamarix shrubs.**

## Effects of soil physical and chemical factors on soil microbial community structure in Tamarix shrubs

In VIF (Variance Inflation Factor) analysis, environmental factors were screened to filter out the factors with VIF value > 20. Furthermore, 11 environmental factors, such as pH and SWC, and the top 10 abundant phyla were selected for db-RDA analysis, a distance-based redundancy analysis. db-RDA analysis was employed to elucidate the correlation between bacterial community and environmental factors. In db-RDA analysis, the abscissa represents the first principal component, and the percentage represents the contribution of the first principal component to the sample difference. The ordinate represents the second principal component, and the percentage represents the contribution of the second principal component to the sample difference. Each point in the figure represents a sample. Environmental factors are generally represented by arrows, and the length of the arrow line represents the degree of correlation between an environmental factor and the distribution of community and species. The longer the line is, the greater the correlation is, and vice versa. Horizontal and vertical coordinates contributed 67.04% and 14.68% of differences in bacterial community composition of soil samples (Fig. 6), respectively, accounting for 81.72% of the total variation in variance. According to db-RDA analysis, EC ($R^2$ = 0.62, $P$ = 0.001), AK ($R^2$ = 0.59, $P$ = 0.001), OM ($R^2$ = 0.52, $P$ = 0.002), AP ($R^2$ = 0.43, $P$ = 0.013), and $NH_4^+$ ($R^2$ = 0.37, $P$ = 0.028) were the primary driving factors in community variation and were significantly correlated to bacterial community. As per the outcomes of this analysis, SWC, TK and pH were
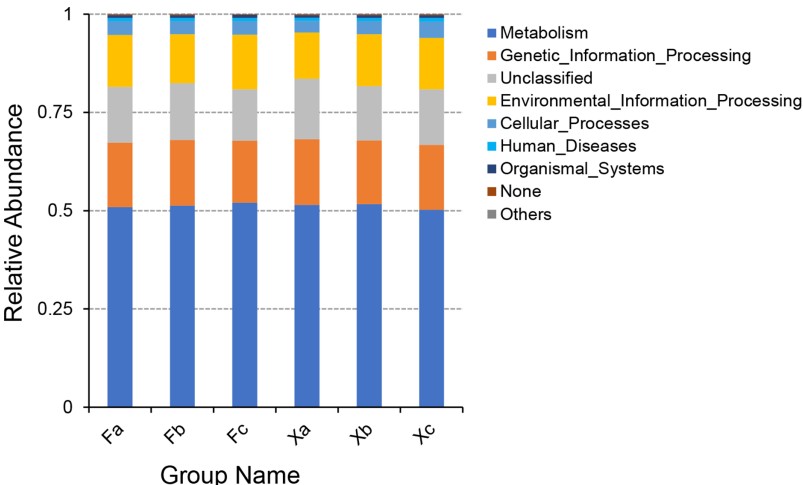

**Figure 7 Column accumulation plot of relative abundance of bacterial community functions in first hierarchy level.**                               

positively correlated with each other, but negatively correlated with other soil factors. Halobacterota was significantly positively correlated with OM and AP ($P < 0.01$), and significantly positively correlated with TN, TP, $NH_4^+$, AK and EC ($P < 0.05$). Also, Bacteroidota was significantly positively correlated with AK ($P < 0.01$) and significantly positively correlated EC ($P < 0.05$), but Chloroflexi had a significant negative correlation with AK and EC ($P < 0.01$). Besides, Proteobacteria was significantly negatively correlated with OM ($P < 0.05$) as well as Actinobacteriota had a significant negative correlation with TN, AK and EC ($P < 0.05$), and Acidobacteriota was significant negatively correlated with TN and EC ($P < 0.05$).

## Gene function analysis of soil bacteria using PICRUSt software in first and second hierarchy level

PICRUSt software was employed to predict the gene function of soil microbiota in Tamarix shrubs based on the annotations in the KEGG database. PICRUSt analysis of soil bacterial genes was performed for gene sequences obtained through high throughput sequencing of soil bacterial, and a total of 6,195 KEGG Orthology (KO) were predicted. As per the outcomes of the functional annotation of these genes using KEGG database, a total of six clear metabolic pathways in first hierarchy level, 41 metabolic pathways in second hierarchy level, and 307 metabolic pathways in third hierarchy level were obtained (Subdividing the first hierarchy level soil bacterial functions to get more second hierarchy level bacterial functions, and subdividing the second hierarchy level bacterial functions to get more third hierarchy level functions by PICRUSt software). The six metabolic pathways in first hierarchy level (Fig. 7) entailed metabolism (51.3%), genetic-information processing (16.4%), environmental information-processing (13%), cellular processes (3.4%), human diseases (0.8%), and organismal systems (0.6%). In addition, the average relative abundance of unclassified and unpredicted (none) functional genes was 14.3% and 0.2%.

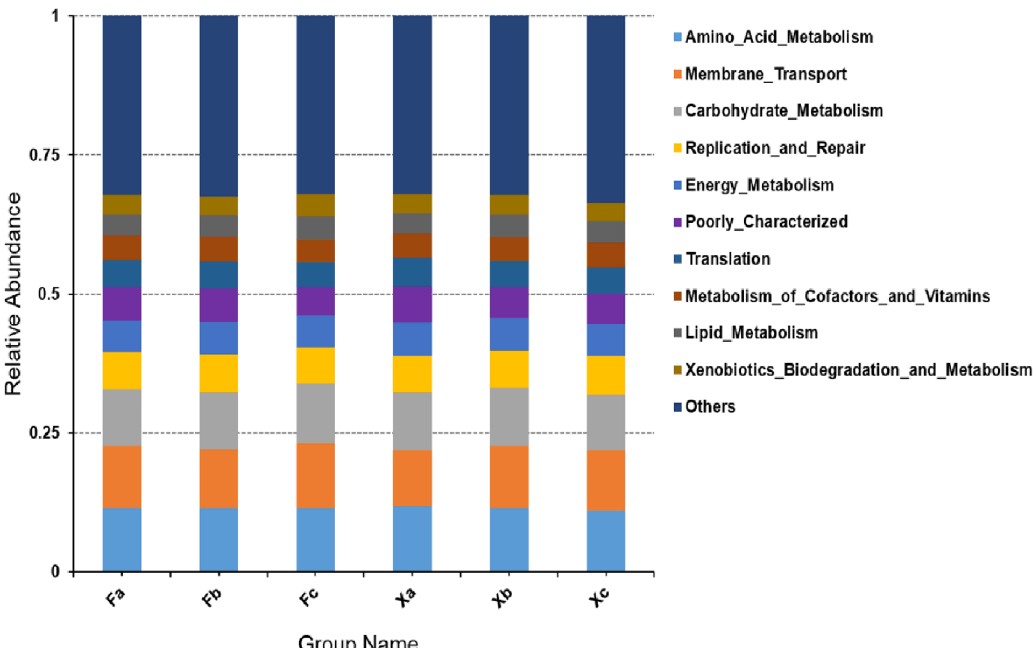

**Figure 8 Column accumulation plot of relative abundance of bacterial community functions in second hierarchy level.**

In second hierarchy level bacterial functions, the top 10 over-represented bacterial functions with the highest abundance were selected to generate a histogram of relative functional abundance (Fig. 8), and these functions included amino-acid metabolism, membrane transport, carbohydrate metabolism, accounting for 11.4%, 10.9%, and 10.3% of total functions respectively. In first hierarchy level soil bacterial functions, the number of metabolic genes occupied a significant advantage. In second hierarchy level bacterial functions, the number of functional genes related to amino acid metabolism and carbohydrate metabolism also occupied a significant advantage. Therefore, it can be inferred that maybe Plenty of the soil bacteria in Tamarix shrubs had active growth and metabolism processes.

## Clustering heat map analysis of soil bacterial function in third hierarchy level

Based on the functional annotation and abundance of the samples, the top 35 over-represented bacterial functions in abundance were selected to construct a cluster heat map in third hierarchy level (Fig. 9). Bacteria with dominant functions almost belonged to Xa, Xb, and Xc groups. We analyzed the dominant functions of bacteria in these three groups. Xa was most distantly related to other groups. It suggested that the dominant bacterial function in Xa group could differ from other groups. In Xa, out of 19 dominant functions of bacteria, 12 functional processes, such as pyruvate-metabolism, citrate cycle (TCA cycle), and methane metabolism, were related to metabolism. In addition, ribosome, transcription-machinery, and DNA replication proteins were associated with genetic information processing. There were seven dominant bacterial functions in Fc, of
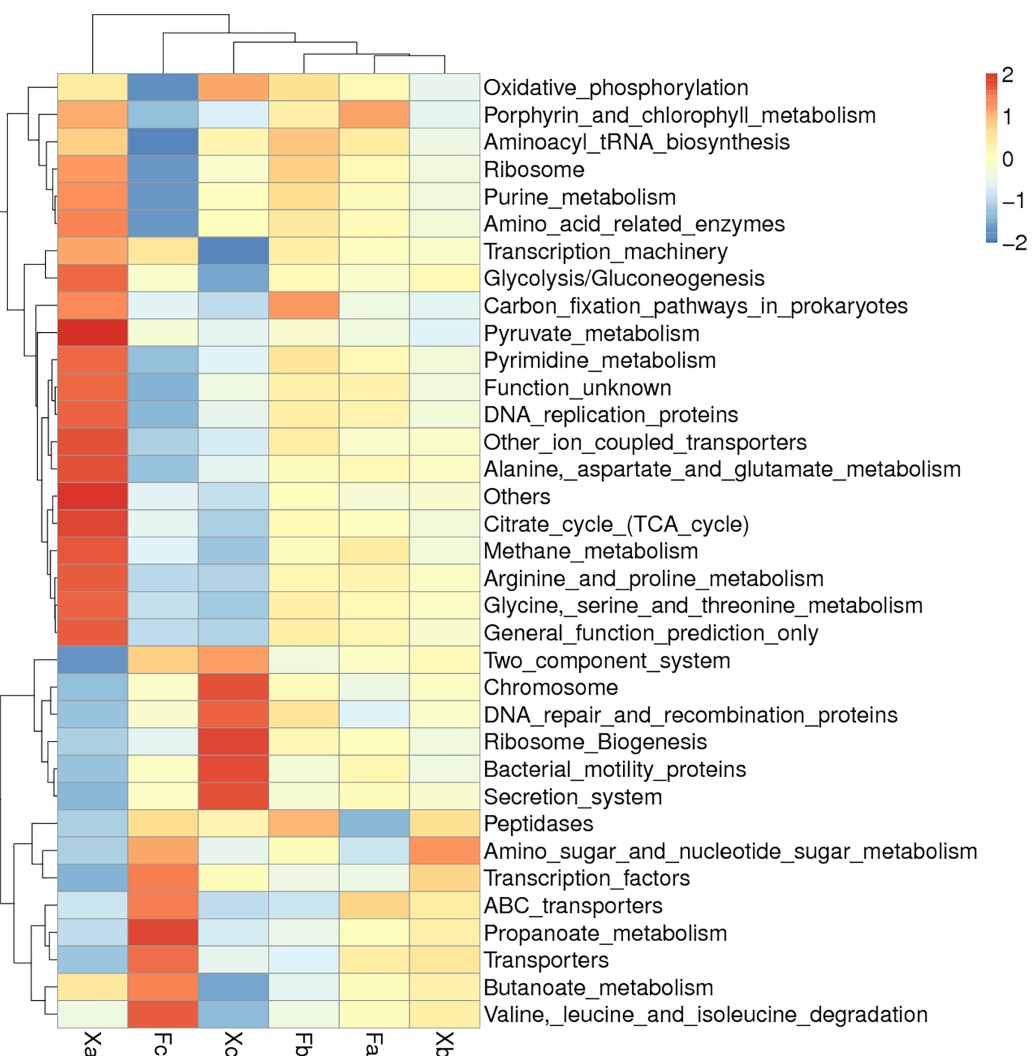

**Figure 9 Clustering heat map of soil bacteria functions in third hierarchy level.**

which four function, *i.e.*, amino-sugar-and-nucleotide-sugar-metabolism, butanoate metabolism, valine-leucine-and-isoleucine-degradation, and propanoate metabolism, were related to metabolism, and two function, namely ABC-transporters and transporters, were related to environmental information processing. Besides, Transcription-factors was related to Genetic Information Processing, Xc group bacteria contributed to a total of seven dominant bacterial functions of which ribosome-biogenesis, chromosome, DNA-repair-and-recombination-proteins are related to genetic information processing. Besides, secretion systems and two-component systems are related to environmental information processing, oxidative phosphorylation is related to metabolism, and bacterial-motility proteins are related to cellular processes. It can be speculated that in *Tamarix* shrubs, the functions related to metabolism in the bacterial community have a dominant advantage, followed by the functions related to genetic information processing and environmental information processing occupy a certain advantage.

## DISCUSSION

Spatial heterogeneity of soil resources is a common phenomenon in arid and semi-arid regions (*Yi et al., 2008*). Apart from this, shrub sand dunes mediated nutrient enrichment is also an important mechanism resulting in soil resources' spatial heterogeneity. The SWC in the study area ranged from 0.65% to 2.84%, and the pH of the soil was 7.5–8.0, and thus soil was classified as the highly arid desert saline-alkali soil. In the current study, apart from pH, TK, and $NH_4^+$, the other soil factors did not differ significantly between nebkhas and non-nebkhas Tamarix shrubs. As stated in the study by *Chen et al. (2015)*, the continuous development and growth of nebkhas shrubs could significantly increase the nebkhas soil's nutrient content. Tamarix nebkhas shrubs selected in the current study were formed recently, and the enrichment effects of soil nutrients was not apparant. In nebkhas and non-nebkhas shrubs, maximum values of OM, TN, TP, AK, EC, and TDS in the canopy suggested that in addition to the effect of the nebkhas shrubs on soil nutrient enrichment, the canopy can also collect nutrients around the soil to a certain extent. This can be attributed to the close proximity of soil in the canopy to the plant roots; also, plant root exudates contain various inorganic ions, sugars, amino acids, and other compounds, which provide nutrients and energy to the surrounding soil (*Tu et al., 2000*). Besides, the litter degradation in the canopy can also improve the soil environmental conditions to a certain extent (*Wang et al., 2020*).

Due to high microbial abundance in the soil, the bacterial community structure of the soil is complex. In this study, a total of two kingdoms, 71 phyla, 161 classes, 345 orders, 473 families, and 702 genera were detected using sequencing technology. Multiple studies have shown that Proteobacteria, Actinobacteriota, Acidobacteriota, Bacteroidota, Firmicutes, and so on are commonly occurring bacteria in the arid desert soil, which is in line with the findings of this study (*Schadt, 2010*; *Osman, Fernandes & DuBow, 2017*). In a study by *Pan (2019)*, Actinobacteria and Proteobacteria were reported to be the dominant bacterial phyla in Nitraria spp. shrubs in the desert steppe. However, the outcomes of the current study were not in line with the study by Pan et al. In this study, we observed that Halobacterota had a significant advantage in Tamarix shrubs. The different geographical environment and vegetation typess ignificantly affects microbiota. The study area belong to a typical arid saline-alkali environment, as well as Tamarix is a typical drought-tolerant and salt-tolerant desert plant. Microbes in Halobacterota can survive in a high salt concentration (*Zhang, 2016*), it has a high salt tolerance and plays a positive role in promoting the salt tolerance of plants (*Wang et al., 2010*). The high salt content in the soil of *Tamarix ramosissima* shrubs was apt for the survival of Halobacterota and had a selection effect on other soil microbes.

Alpha diversity of soil bacterial communities in Tamarix shrubs, calculated using Shannon, Simpson, Chao1, and ACE indices, was not significantly different in nebkhas and non- nebkhas Tamarix shrubs. It indicated that nebkhas had an insignificant effect on soil bacterial community diversity and abundances. It may be due to no apparent differences in soil factors within the Tamarix shrubs. A total of 63% of the total OTUs were shared by nebkhas and non-nebkhas Tamarix shrubs. It indicated that nebkhas and

non-nebkhas Tamarix shrubs contained significantly overlapping bacterial communities. OTUs in the canopy's inner and edge were greater than OTUs in the edge of the shrubs. Besides, OTUs in nebkhas Tamarix shrubs were higher than non-nebkhas Tamarix shrubs. This indicated that Tamarix canopy and Tamarix nebkhas had a certain enrichment effect on soil bacteria.

NMDS analysis and UPGMA cluster analysis demonstrated certain differences in soil bacterial community structure between nebkhas and non-nebkhas Tamarix shrubs. The difference of bacterial community structure in Tamarix nebkhas shrubs was higher than that in the non-nebkhas shrubs. Besides, the soil bacterial community composition and structure of Xa was some difference than other groups (Figs. 3, 5), and LEFSe analysis revealed most biomarker in Xa (Fig. 4). These findings showed that the species composition and community structure of soil bacterial community in Xa was visible different from other groups. It indicated that Tamarix nebkhas and vegetation together had obvious effection on the structure and composition of the soil microbial community.

The correlation between soil bacterial community structure and soil physicochemical factors was elucidated through correlation analysis. As per the outcomes of this analysis, OM, AK, AP, EC, and $NH_4^+$ were found to be the primary soil factors affecting soil bacterial community structure. It indicated that soil nutrients and salt had significant effects on the bacterial community. *Dai et al. (2017)* investigated the soil bacterial community of different vegetation types in the Hobq Sand and reported that OM, SWC, AN, AP, and TN mainly affected the abundance and diversity of soil bacterial community. In addition, *Niu et al. (2017)* investigated the microbial diversity in saline-alkali soil in the Hexi Corridor and reported that the soil pH, OM, TN, and EC significantly affected the microbial community composition of this ecological site. Thus, it can be inferred that the physical and chemical properties of soil affect the bacterial community structure present in different types of desert saline-alkali soil. In db-RDA analysis, we observed that Halobacterota is significantly positively correlated with EC, AP, AK, $NH_4^+$ and OM. Halobacterota, the dominant bacteria phylum in Tamarix shrubs, is a halophilic heterotrophic microbe with high salt tolerance. These microbes can survive in an environment of high salt concentration (*Wang et al., 2010*). AP, AK, $NH_4^+$ and OM serve as available nutrient and energy sources, which meet the energy requirements pertaining to metabolic processes in Halobacterota and imparts "salt tolerance" or, in other words, promotes a "salt-loving" lifestyle in microbes inhabiting saline-alkaline environments (*Zhang & Fan, 2002*).

Through bacterial gene function prediction, we observed that metabolism-related functional genes were dominant in all three hierarchy levels, followed by the gene information processing genes; besides, environmental information processing related functional bacteria were found to have significant advantages. The relative abundance of the various functional bacteria in first and second hierarchy levels was more evenly distributed in each group (Figs. 7, 8). It indicated that the spatial heterogeneity of the bacterial functional genes of Tamarix shrubs is small. In second hierarchy level, the

number of functional genes related to amino acid metabolism, membrane transport, and carbohydrate metabolism was significantly dominant, in line with the previous arid environment-related (*Sun et al., 2020*) and saline-alkali land literature (*Tang et al., 2020*). It indicated that active growth and metabolism activities of bacteria are common in the arid saline-alkali environments. Carbohydrate metabolism plays a crucial role in the biochemical process by regulating the formation, decomposition, and mutual transformation of carbohydrates in microbes. Plants and microbes absorb ammonium salts, nitrates, and other inorganic nitrogen from the environment and synthesize proteins and nitrogen-containing substances through amino acid metabolism (*Miflin & Lea, 1977*). Drought stress leads to the accumulation of multiple substances regulating osmotic pressure in the cells. The concentration of these substances increases in the cytoplasm, thereby decreasing the osmotic potential of the cell, which is regulated by chemical signals transmitted through membrane transport (*Bhattacharya, 2010*). Therefore, the above three functional bacteria's enrichment indicates that the microbiota might improve the surrounding soil nutrients, provide nutrients for plants, and promote plant growth under drought stress in arid desert environments.

## CONCLUSION

In this study, Tamarix nebkhas and Tamarix canopy showed a certain enrichment effect on soil nutrients, content, and bacterial community structure. The comprehensive effect of nebkhas and canopy had obvious effects on the structure and function of the soil bacterial community. The number of Tamarix shrubs OTUs annotated to the database accounted for 99.8% of total OTUs. At different classification levels, a total of two kingdoms, 71 phyla, 161 classes, 345 orders, 473 families, and 702 genera were detected. At the phylum level, Halobacterota, unidentified bacteria, and Proteobacteria were identified to be the dominant phyla of Tamarix shrubs. OM, AK, AP, EC, and $NH_4^+$ were the primary soil factors to affect the structure of the soil bacterial community. SWC, TK and pH were positively correlated with each other, but negatively correlated with other soil factors. At the phylum level, a significantly positive correlation was observed between the Halobacterota and AP, OM as well as Bacteroidota and AK ($P < 0.01$), but a significantly negative correlation was observed between the Chloroflexi and AK, EC ($P < 0.01$). A total of 6,195 KO was detected in Tamarix shrubs, and six metabolic pathways in first hierarchy level, 41 metabolic pathways in second hierarchy level, and 307 metabolic pathways in third hierarchylevel were obtained by functional annotation of sequences from soil bacteria. Among them, the functional genes related to metabolism, gene information processing, and environmental information processing were dominant.

As the nebkhas shrubs selected in this experiment were formed recently, the enrichment of soil nutrients and soil bacteria by nebkhas was not obvious. Two aspects-the nebkhas shrubs with a long time of formation and soil depths in nebkhas shrubs can be selected to conduct in-depth research on soil microorganisms to provide more basic and scientific data for the ecological restoration of Tarim River in the future.

### Funding

This work was supported by the National Natural Science Foundation of China (No. 41561010). The funders had no role in study design, data collection and analysis, decision to publish, or preparation of the manuscript.

### Grant Disclosures

The following grant information was disclosed by the authors:
National Natural Science Foundation of China: 41561010.

### Competing Interests

The authors declare that they have no competing interests.

### Author Contributions

- Fangnan Xiao conceived and designed the experiments, performed the experiments, analyzed the data, prepared figures and/or tables, authored or reviewed drafts of the paper, and approved the final draft.
- Yuanyuan Li conceived and designed the experiments, performed the experiments, prepared figures and/or tables, authored or reviewed drafts of the paper, and approved the final draft.
- Guifang Li analyzed the data, prepared figures and/or tables, authored or reviewed drafts of the paper, and approved the final draft.
- Yaling He conceived and designed the experiments, performed the experiments, authored or reviewed drafts of the paper, and approved the final draft.
- Xinhua Lv performed the experiments, analyzed the data, authored or reviewed drafts of the paper, and approved the final draft.
- Li Zhuang conceived and designed the experiments, analyzed the data, prepared figures and/or tables, authored or reviewed drafts of the paper, and approved the final draft.
- Xiaozhen Pu conceived and designed the experiments, performed the experiments, analyzed the data, prepared figures and/or tables, authored or reviewed drafts of the paper, and approved the final draft.

### Data Availability

   The data is available at NCBI SRA: PRJNA726047.

### Supplemental Information

Supplemental information for this article can be found online at http://dx.doi.org/10.7717/peerj.12105#supplemental-information.

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
