# Peer review of "High throughput sequencing-based analysis of the soil bacterial community structure and functions of Tamarix shrubs in the lower reaches of the Tarim River"

_PeerJ, doi:10.7717/peerj.12105_

## Round 0.1 · original submission · Minor Revisions

Dear authors:

Thank you for your paper. After revising the content of the manuscript and the concerns raised by the reviewers I kindly request that you address all the aspects and concerns that the reviewers had. One particular concern that I have myself is why not control samples for the analysis of the microbial community composition not associated with the shrubs was done? Please clarify this aspect.

Also, I strongly recommend a second revision of the writing to ensure that all aspects of the writing are correct.

I think that the study is of value and would add a great manuscript to PeerJ. So I look forward to your revisions

Kind regards

·

Basic reporting

no comment

Experimental design

This in an interesting report, evaluating the microbial communities associated to Tamarix shrubs found in nekhabs dunes and in no dunes, and although the experimental procedures are well explained and in my opinion valid, I have some questions:

1) Why not control samples for the analysis of the microbial community composition not associated with the shrubs was done? is it enough with the samples taken from the edges of the shrubs and canopies?

2) The weather of the sampling site (temperature, precipitation, etc) was very variable, when exactly were the samples taken? Do you expect significant changes in microbial composition in different seasons?

3) although the microbial communities in shrubs of similar sizes were analyzed are there morphological differences between the ones in nekhabs and the ones in no nekhabs?

Validity of the findings

I think the findings are valid, although I have some doubts about the experimental design.

Additional comments

it is an interesting and informative work, but I have some concerns already stated, related to the experimental design and also some minor points:

1) abstract mentioning "very significnatly" possitive or negative correlation is not necessary, better is to mention the p values.

2) in several parts of the text there are extra spaces between words and some words with no spaces between them. e.g. L 36 "well as Bacteroidota", L 263 "analysis(Figure 5).", etc, please correct this and "justify" the text.

3) L 313 "human diseases (0.8%)," do you mean genes encoding virulence factors?

4) L. 341 "functiona"?

5) L 407 "had obvious affection" what do you mean?

6) more informative figure legends are needed.

Reviewer 2 ·

Basic reporting

no comment

Experimental design

no comment

Validity of the findings

An important problem I encounter is whether there are differences in the microbial communitiesin the nebkhas and no-nebkhas Tamarix shrubs, the results (269) shown ""It indicated that the diversity of bacterial community structure in nebkhas Tamarix shrubs was greater than that in non-nebkhas Tamarix shrubs" but in the discussion says (389)"Alpha diversity of soil bacterial communities in Tamarix shrubs, calculated using Shannon,Simpson, Chao1, and ACE indices, was not significantly different in nebkhas and non- nebkhas Tamarix shrubs. It indicated that nebkhas had an insignificant effect on soil bacterial community diversity and abundances" However in Halobacterota there are differences (figure 3).is enough to say that the microbial communities are different in nebkhas and no-nebkhas Tamarix shrubs?

Additional comments

This study describes the results of massive sequencing and physicochemical characteristics of two types of soils in the lower Tarim river basin in China, particularly in a type of soil called nebkhas which contains shrubs of a genus called Tamarix, This paper can be published in Peer J as an article after solving some questions and correcting some typos.

106: for a clearer understanding for the reader, it would be desirable to visualize the distinction between "the nebkhas (X) and three non-nebkhas Tamarix shrubs (F)", as shown in the figure 2 from paper: Plant Soil (2011) 347:79–90 or in the figure 4 from GEOMATICS, NATURAL HAZARDS AND RISK 2019, VOL. 10, NO. 1, 1176–1192.

by way of the figure legends could be more detailed for a better understanding of the figures.

In table 1 and 2 it is confusing to determine the statistical significance with the nomenclature proposed by the authors, I suggest an additional table showing the statistical significance.

179: the Authors say: "The pH of nebkhas Tamarix shrubs was significantly higher than that of non-nebkhas Tamarix shrubs" the values are 7.88 vs 7.59", that values are statistical significant, Ok, but, has this difference some biological consequence?

185,187 : similar to the case of pH, the values ​​of TK and NH4+ were significantly different in nebkhas and no-nebkhas tamarix shrubs,has difference some biological consequence? or is only statistical

257: the analysis NMDS is not clear, What does it mean in this analysis? What does it mean "NMDS can analyze the differences between samples in low dimensions" and define the term "bacterial community structure" because in 266 :These results indicated differences in bacterial community structure between nebkhas and non-nebkhas Tamarix shrubs" what does it mean?

326: please clarify, how to explain?: "Therefore, it can be inferred that most of the soil bacteria in Tamarix shrubs had active growth and metabolism processes", Finding more genes related to metabolism does not mean that there is an active growth in these soils, it is possible that in the genomes of these bacteria this group of genes is just overrepresented.


360 and 468: how was determined that Tamarix nebkhas shrubs formed recently?

380 :could you please name some members of this plhylum Halobacterota?


typo: table 1 PH is pH

233: typo: FC is Fc

245:explicitly define LDA


248: typo: FC is Fc

338: typo: FC is Fc

356: typo: PH is pH

---

## Round 0.2 · accepted · Accept

Thank you so much for addressing all the comments made by the reviewers. I hope your paper gets many citations!

Warm regards

·

Basic reporting

it is OK

Experimental design

it is OK

Validity of the findings

it is OK

Additional comments

Thanks for addressing my comments!

Reviewer 2 ·

Basic reporting

no comment

Experimental design

no comment

Validity of the findings

no comment